# Hypercapnia in COPD: Causes, Consequences, and Therapy

**DOI:** 10.3390/jcm11113180

**Published:** 2022-06-02

**Authors:** Balázs Csoma, Maria Rosaria Vulpi, Silvano Dragonieri, Andrew Bentley, Timothy Felton, Zsófia Lázár, Andras Bikov

**Affiliations:** 1Department of Pulmonology, Semmelweis University, 25-29 Tömő Str., 1083 Budapest, Hungary; csoma.balazs@semmelweis-univ.hu (B.C.); lazar.zsofia@med.semmelweis-univ.hu (Z.L.); 2School of Medicine: Basic Medical Sciences, Neuroscience and Sense Organs, University of Bari Aldo Moro, 11 Piazza G. Cesare-Bari, 70124 Bari, Italy; mariarosaria.vulpi@gmail.com (M.R.V.); silvano.dragonieri@uniba.it (S.D.); 3Wythenshawe Hospital, Manchester University NHS Foundation Trust, Southmoor Road, Manchester M23 9LT, UK; andrew.bentley@mft.nhs.uk (A.B.); tim.felton@manchester.ac.uk (T.F.)

**Keywords:** airway immunity, chronic obstructive pulmonary disease, hypercapnia, noninvasive ventilation

## Abstract

Chronic obstructive pulmonary disease (COPD) is a progressive respiratory disorder that may lead to gas exchange abnormalities, including hypercapnia. Chronic hypercapnia is an independent risk factor of mortality in COPD, leading to epithelial dysfunction and impaired lung immunity. Moreover, chronic hypercapnia affects the cardiovascular physiology, increases the risk of cardiovascular morbidity and mortality, and promotes muscle wasting and musculoskeletal abnormalities. Noninvasive ventilation is a widely used technique to remove carbon dioxide, and several studies have investigated its role in COPD. In the present review, we aim to summarize the causes and effects of chronic hypercapnia in COPD. Furthermore, we discuss the use of domiciliary noninvasive ventilation as a treatment option for hypercapnia while highlighting the controversies within the evidence. Finally, we provide some insightful clinical recommendations and draw attention to possible future research areas.

## 1. Introduction

Chronic obstructive pulmonary disease (COPD) is a preventable and treatable disease that is caused by aberrant airway inflammation due to chronic exposure to noxious gases or particles, such as tobacco or biomass fuel smoke [1]. COPD is responsible for 250,000 deaths yearly in the European Union and is the third leading cause of mortality worldwide [2]. It is usually considered to be a progressive disease, and the ultimate worsening of airflow limitation and impaired gas exchange may result in the development of hypercapnia [3]. The presence of alveolar hypoventilation, and therefore the development of consequential hypercapnia, is more common in severe forms of COPD [4], although hypercapnia is present in even moderate disease in a proportion of patients [5]. The prevalence of hypercapnia is around 30–50% in patients with very severe COPD (predicted forced expiratory volume in the first second (FEV_1_) <30%) [5]. Although hypercapnia may be beneficial to mitigate pulmonary inflammation [6], there is an increasing amount of evidence suggesting that the deleterious effects outweigh the protective ones [7,8,9,10]. Hypercapnia causes alveolar epithelial dysfunction, which results in alveolar oedema formation and further deterioration in gas exchange [7,11]. Moreover, the repair mechanisms of the airway epithelial cells are also damaged as hypercapnia causes mitochondrial dysfunction and impaired cell proliferation [12]. Overall, chronic hypercapnia is an independent risk factor for hospitalizations and mortality in COPD and has thus attracted attention as a possible treatable trait [10,13,14,15,16].

Several mechanisms, which are detailed in this review article, may lead to alveolar hypoventilation and result in hypercapnia in COPD. Correction of hypoventilation and acute respiratory acidosis using noninvasive ventilation (NIV) is a widely used and evidence-based treatment for acute hypercapnic respiratory failure (AHRF) [17]. It is also available at domiciliary settings for patients with chronic type II respiratory failure (i.e., hypoxemia with hypercapnia), including those with COPD [18]. However, the effects of long-term NIV (LT-NIV) on hard endpoints (mortality and exacerbation frequency) are unclear due to the contradictory results of previous studies in COPD [18,19].

The mechanisms leading to hypercapnia in COPD are well documented, and we aim to summarize them only briefly. Instead, our review will focus on the pulmonary and systemic effects of hypercapnia and evidence on the effects of long-term NIV treatment in COPD. We highlight controversies, provide clinical recommendations, and outline an agenda for further research.

## 2. The Mechanisms of Hypercapnia in Stable COPD

As COPD is a heterogenous disorder, it is not surprising that mechanisms leading to hypercapnia are also numerous and not necessarily interrelated. Identification of these processes as treatable traits is important at the individual level. The following risk factors are described to be associated with the development of chronic hypercapnia in patients with COPD: breathing pattern, inspiratory muscle weakness, smoking habit, low FEV_1_, high body mass index, reduced forced vital capacity, and high arterial bicarbonate level [3,20,21,22].

Arterial carbon dioxide pressure (PaCO_2_) is determined by the ratio of CO_2_ production and alveolar ventilation. Although CO_2_ production may be accelerated due to hypoxemia and sarcopenia, the predominant process leading to hypercapnia is alveolar hypoventilation [23]. Alveolar ventilation is determined by two main processes: minute ventilation and dead space ventilation [24]. Processes leading to decreased respiratory drive, including reduced central nervous system control (“the patient will not breathe”) and impaired respiratory muscle strength (“the patient cannot breathe”), decrease minute ventilation, whereas increased dead space area due to airway, parenchymal, and vascular changes in COPD promotes dead space ventilation. Most importantly, hypercapnia occurs when a patient is unable to sustain satisfactory ventilation due to overload of the respiratory system.

The “overload” that challenges the thoracopulmonary system is due to increased resistance of the airways, pulmonary hyperinflation, and increased ventilatory demand [25,26]. Pulmonary hyperinflation plays a central role in the pathogenesis of hypercapnia in COPD [27], especially in severe disease [28]. Static pulmonary hyperinflation, i.e., the increase in functional residual capacity and the ratio of residual volume/total lung capacity (RV/TLC) at rest, causes the thoracopulmonary system to breathe at rest at higher volumes during full expiration on a portion of the most unfavorable length–tension curve [29] (Figure 1). This “disadvantage” causes muscle fibers to adapt by shortening diaphragmatic sarcomeres and increasing the proportion of type I fibers. Despite the presence of these compensatory mechanisms, pulmonary hyperinflation impairs ventilatory capacity in patients with COPD, with the system unable to increase both tidal volume and expiratory volume when ventilatory demand increases as it occurs during exercise or exacerbation. Furthermore, hyperinflation decreases the resting length of the diaphragm and the rib cage muscles [30,31], causing diaphragmatic dysfunction with a significant increase in respiratory work at rest.

This “overloaded system” is superimposed on an unfavorable muscular condition that is unable to support this increased workload [32,33,34]. The unfavorable alterations in the muscles are also due to compromised oxygen delivery, the nutritional deficit that typically occurs in patients with severe COPD and systemic inflammation [34,35]. However, not all authors agree on defining the muscular system of hypercapnic COPD patients as “weaker”. Topeli et al. showed that there is evidence of greater activation of the diaphragm following increased ventilatory demand in hypercapnic COPD patients compared to normocapnic COPD patients [36].

Due to progressive alveolar tissue loss, the physiological ventilation/perfusion (V/Q) imbalance can be accelerated in patients with COPD. The alveolar tissue loss is usually heterogenous and leads to poorly ventilated but well perfused (lower lobes) or well ventilated but poorly perfused (upper lobes) areas, thereby increasing the physiological dead space areas [37]. An increase in dead space ventilation naturally leads to decreased alveolar ventilation. The V/Q can further be worsened by the presence of comorbidities (e.g., coagulopathy, heart disease, and pulmonary hypertension). The central respiratory drive can also be reduced in a number of patients with COPD [38]. The control of ventilation depends on the interaction of signals from central chemoreceptors and pressure receptors on the rib cage and respiratory muscles. Moreover, downregulation of the breathing centers, with the consequent alveolar hypoventilation, seems to be a protective factor put in place to “save energy” and avoid additional burden on the respiratory system [39,40].

Usually, hypoxemia precedes hypercapnia in patients with COPD. Long-term O_2_ treatment (LTOT) is beneficial in these patients; however, it may induce hypercapnia in otherwise normocapnic individuals [41], and routine blood gas analyses are therefore recommended in these patients.

In addition, COPD may be accompanied by other disorders associated with hypercapnia, such as obesity [42], chest wall diseases [43], and neuromuscular disease [44]. Of note, obesity may also occur due to immobility commonly seen in patients with COPD. Recognizing them during the diagnostic workup is essential when tailoring individualized treatment.

## 3. The Mechanisms of Hypercapnia during Exacerbation of COPD

Many COPD exacerbations are characterized by hypercapnic hypoxemic respiratory failure, which is a major risk factor for mortality [45,46]. Acute hypercapnia can be found both in patients who are already hypercapnic at rest and in patients with normocapnia, in whom exacerbation imposes an excessive load on the thoracopulmonary system that leads to difficulty in sustaining the ventilatory flow. A history of severe exacerbations in previous years is the most important risk factor for the occurrence of future flare-ups [47,48,49]. Furthermore, developing hypercapnia during an exacerbation correlates with the risk of mortality in the following 12 months [50], thus representing a negative prognostic factor.

Alterations in the V/Q ratio represent one of the first and main mechanisms that occurs during COPD exacerbation and leads to the development of hypercapnic respiratory failure [51,52]. Many regions of the parenchyma are perfused but not ventilated due to the presence of bronchospasm, edema, and secretions [53], which increase dead space ventilation. In addition, there is an increase in respiratory and muscular work, with a consequent rise in oxygen demand [54]. Barbera et al. showed that hypercapnic respiratory failure during COPD exacerbation is mainly linked to alterations in V/Q as a result of remodeling of the airways, bronchospasm, and hypersecretions [55].

The establishment of dynamic pulmonary hyperinflation, i.e., the increase in end-expiratory lung volume (EELV) above relaxation volume [56], can contribute to an increase in PaCO_2_ during exacerbations, which also develops in patients with flow limitation during exercise. During exacerbations, the resistance of the airways, and therefore the flow restriction, increases due to bronchoconstriction, edema of the walls, and accumulation of bronchial secretions [57]. This is superimposed on hyperactivation of the respiratory neural drive with rapid shallow breathing pattern and consequent lengthening of the time constant and air entrapment [57]. These elements cause a progressive vicious circle underlying dyspnea, a key symptom of COPD exacerbations. As a “mechanical” consequence, patients find themselves breathing in an even more unfavorable portion of the compliance curve, in which large changes in pressure cause small changes in volume with a consequent increase in respiratory work.

The abovementioned factors may lead to the presence of positive static end-expiratory elastic recoil pressure called intrinsic positive end expiratory pressure (PEEP), which represents an additional mechanical load that the respiratory system must overcome in order to guarantee respiratory gas exchanges [58]. This implies that the respiratory muscles must first overcome the intrinsic PEEP threshold in order to induce an adequate inspiratory flow. Furthermore, regional differences in intrinsic PEEP contribute to poor pulmonary ventilation distribution and thus to impaired gas exchange.

All these elements contribute to create an additional load on the already compromised thoracopulmonary system, with consequent muscle weakness and overload, particularly at the diaphragmatic level [59]. In this way, the work necessary to ensure respiratory exchanges is significantly increased, with hyperactivation of the central respiratory tracts that does not correspond to a proportional increase in muscle activation due to fatigue [57].

An option to balance the load and capacity of the ventilatory muscles is the use of mechanical ventilation. Positive-pressure support ventilation unloads fatigued respiratory muscles, thus enabling recovery of the respiratory system and leading to improvement in lung function parameters, correction of hypercapnia, and reversal of acidosis [60]. The main types of ventilatory support are invasive and noninvasive ventilation. According to current guidelines, due to its several advantages over invasive ventilation (no need for sedation; maintained ability to eat, drink, cough, and expectorate; and lower risk of ventilator-associated pneumonia) in the rightly selected population, NIV is the recommended initial therapy for acute hypercapnic respiratory failure during an acute exacerbation of COPD [61].

The mechanisms leading to hypercapnia in stable and exacerbated conditions are summarized in Figure 2.

## 4. Effects of Hypercapnia on the Lung

Carbon dioxide is not only a by-product of physiological processes but also a signaling molecule, and it can affect many different cell and tissue types [62]. The pathways through which CO_2_ impacts cell functions include pH-dependent and pH-independent routes via soluble and transmembrane receptor proteins [63,64,65]. In the respiratory system, hypercapnia affects the alveolar epithelial function and cell repair mechanisms, the inflammatory response and immunity of the airways, and the airway mechanics [66].

### 4.1. Effects on Alveolar Epithelium

The most thoroughly studied effect of hypercapnia on the respiratory system is its relationship with alveolar epithelial dysfunction. The deleterious effects of hypercapnia are multifactorial. Firstly, in vitro and rodent models have been used to show that resorption of the alveolar fluid is impaired due to endocytosis of Na^+^/K^+^-ATPase enzyme, leading to reduced clearance of lung oedema. The reduction of Na^+^/K^+^-ATPase activity is mediated through protein kinase C and A pathways [7,64,67]. In addition, it has been reported that hypercapnia attenuates epithelial cell repair by disruption of cell migration via reduced NF-κB activation [68]. The repair mechanism is further impaired through disturbed plasma membrane wound resealing [69,70] and decreased rate of proliferation of alveolar epithelial cells due to mitochondrial dysfunction [12].

### 4.2. Effects on Immunity and Inflammatory Response of the Respiratory System

Hypercapnia also affects the immunity of the airways. Although the elevation of CO_2_ levels alters the expression of several hundred genes [71], the key regulator of the response to hypercapnia may be the NF-κB pathway [72]. Several studies have reported that hypercapnia hinders the activation of NF-κB, which regulates genes involved in innate immunity and inflammation [73,74,75]. The consequence of NF-κB suppression is a complex disturbance in the innate immunity, which involves impaired phagocytic capacity of alveolar macrophages and decreased production of proinflammatory cytokines, such as interleukin-6 and tumor necrosis factor-α [9,76]. Moreover, the antiviral activity of macrophages is also impaired in hypercapnia. In a combined in vivo and in vitro model, Casalino-Matsuda et al. showed that hypercapnia increased the replication of influenza A virus in mice while inhibiting the antiviral gene and protein expression in macrophages through activation of the Akt1 pathway [77]. Furthermore, in a recent study, gene clusters that are associated with innate immunity and nucleosome assembly were found to be downregulated by hypercapnia, whereas lipid metabolism genes were upregulated [71].

Besides in vitro models, the effects of hypercapnia on host defense can also be observed in preclinical rodent models of acute lung injury, although the results are controversial. Chonghaile et al. reported that hypercapnic acidosis may attenuate the lung injury induced by *E. coli* pneumonia but only when CO_2_ is administered with antibiotic therapy [78]. Masterson and her colleagues also found that hypercapnic acidosis improved survival and reduced lung inflammation in rats and that the mechanism behind this effect was the inhibition of NF-κB [75]. On the contrary, O’Croinin observed that hypercapnia did not have a protective effect against pneumonia-induced lung injury [79] or even worsened the damage [80].

### 4.3. Effects on Airway Mechanics

Hypercapnia regulates the bronchial muscle tone through vagal nerve stimulation, CO_2_-signalling pathways, and pH-dependent mechanisms. However, the results are controversial as several studies have reported increased airway contractility [81,82,83] while others found that hypercapnia augments airway relaxation [84,85,86,87,88,89,90].

Historically, the dominant explanation of the bronchoconstrictor effect of hypercapnia had been that CO_2_ stimulates vagal efferent nerves, which play an important role in regulating bronchial tone. Later, however, it was proven that the constrictor effect cannot be entirely antagonized by the blockage of the vagal nerve, and other factors also contribute to the observed effect of CO_2_ [65]. Subsequently, in a mouse and human airway smooth muscle cell culture model, Shigemura et al. found that hypercapnia increased acetylcholine-mediated cell contraction, and the effect was independent of oxygen saturation and extracellular pH [81]. Importantly, they validated the in vitro results in a small cohort of human subjects with COPD. It was found that hypercapnic COPD patients had higher airway resistance, which decreased after the normalization of CO_2_ levels [81]. In line with this, using a forced oscillation technique, Uno and his colleagues found that respiratory resistance was higher in hypercapnia patients with COPD [83]. These findings indicate that hypercapnia may contribute to airflow limitation, which severely impacts the quality of life and mortality, particularly in COPD [91,92]. Indeed, randomized controlled trials that aimed to correct chronic hypercapnia in COPD have reported an improvement in lung function parameters, health-related quality of life, and mortality [93,94].

Hypercapnia can also be associated with bronchodilation [84,85,86,87,88,89,90]. In vivo animal models have shown that the inhalation of CO_2_ can reverse the bronchoconstrictor effect of drugs or pulmonary artery occlusion [85,87,90]. A possible explanation for this effect is that carbon dioxide rapidly dissolves into H^+^ and HCO_3_^−^ in a watery solution, thus lowering the intracellular pH in smooth muscle cells. The intracellular acidosis then reduces the influx of ionized calcium (Ca^2+^) through voltage-dependent Ca^2+^ channels, which would be necessary for the active contraction of the muscle [86,88,89]. Moreover, El Mays et al. described a pH-independent, epithelium-mediated mechanism of bronchial relaxation induced by CO_2_. In a rodent in vitro model, they found that substance P provoked relaxation in bronchial rings pretreated with methacholine independently of extracellular H^+^ concentration [84]. Overall, the relaxing effect of hypercapnia seems to be significant only on an already pathologically constricted smooth muscle.

### 4.4. Effects on Pulmonary Circulation

Hypercapnia has also been linked to increased pulmonary vascular resistance and the development of pulmonary hypertension (PH) in COPD. It is estimated that 1–4% of patients with COPD present with severe pulmonary hypertension as defined by mean pulmonary arterial pressure (PAP) ≥ 35 mmHg [95,96]. Of clinical relevance, it has recently been shown that severe PH is an independent risk factor of mortality in COPD [97].

Several lines of evidence confirm the role of hypercapnia in the development of PH in COPD. In an animal model, it was shown that chronic intermittent hypoxia with hypercapnia resulted in elevated pulmonary and systemic arterial pressures and an increase in hematocrit [98]. Importantly, PH in end-stage COPD is not only predicted by hypoxemia but also by hypercapnia [96]. Furthermore, in experimental settings, the development of hypercapnia is associated with an increase in pulmonary vascular resistance [99,100], impaired right ventricular function [100,101], and reduced right ventricular ejection fraction [100]. Hypercapnia can lead to an increase in PAP via acidosis [102], and chronic hypercapnic acidosis is suspected to contribute to heart failure in PH [103]. Importantly, in a small cohort of patients with PH related to hypoventilation due to COPD alone or in combination with obesity hyperventilation syndrome, NIV reduced PAP and improved exercise capacity and cardiac function as assessed by serum N-terminal probrain natriuretic peptide levels [104]. The positive relationship between PAP and body mass index and night-time CO_2_ tension, but not oxygen tension, lung volumes, or left ventricular function, suggests that PH in this population is driven by body mass index and hypercapnic pulmonary vasoconstriction [104].

## 5. Systemic Effects of Hypercapnia

As hypercapnia is a systemic condition, it affects all parts of the human body. Evidence suggests a significant role of carbon dioxide in regulating coronary perfusion as well as muscle function and contractility [105,106,107].

### 5.1. Cardiovascular Effects

Cardiovascular diseases often accompany COPD and impact COPD-related morbidity and mortality [108]. Therefore, cardiovascular consequences of hypercapnia are worth acknowledging.

Elevated PaCO_2_ and the consequent change in pH have regulatory effects on coronary blood flow and oxygenation [105,109]. Hypercapnia significantly increased myocardial blood flow in healthy subjects assessed with positron emission tomography [110,111]. This effect is driven by the direct vasodilating effect of CO_2_ and decreased pH as well as changes in the activity of the autonomic nervous system mediated by central and peripheral chemoreceptor stimulation [105]. Furthermore, hypercapnia promotes improvement in tissue oxygenation through the rightward shift of the oxyhemoglobin dissociation curve, which decreases the affinity of hemoglobin to oxygen, thereby enhancing cellular O_2_ diffusion and uptake (“Bohr effect”) [112]. In addition, elevated CO_2_ tension potentiates hypoxia-induced pulmonary vasoconstriction, thus reducing intrapulmonary shunt and improving the V/Q ratio [113,114]. Besides enhanced oxygenation and coronary perfusion, hypercapnia decreases the total peripheral resistance, consequently lowering the afterload of the left heart and increasing the cardiac output and cardiac index [115,116].

### 5.2. Musculoskeletal Effects

Muscle wasting and skeletal muscle dysfunction are common features of COPD [117]. Muscle loss is independently associated with unfavorable outcomes, such as higher mortality and hospital readmission rates and lower quality of life [118,119,120]. However, the exact link between hypercapnia and muscle dysfunction has been demonstrated only in recent years [117]. Jaitovich and his colleagues studied mice and cultured myotubes that were exposed to high CO_2_ [121]. They observed that the muscle atrophy and reduced anabolic capacity was mediated through AMP-activated kinase (AMPK) subunit α2 activation, which led to an upregulation of muscle-specific ring finger protein-1 (MuRF-1) expression. MuRF-1 is a specific type of ubiquitin ligase responsible for protein degradation [122]. The same research group also showed that AMPKα2 downregulated ribosomal biogenesis and thus decreased muscle protein synthesis, further negatively influencing the skeletal muscle turnover [106]. Furthermore, the regeneration of myoblasts is also impaired due to increased fatty acid oxidation caused by an enhanced rate of oxidative phosphorylation [8].

The possible cellular effects of hypercapnia in patients with COPD are shown in Figure 3.

## 6. Noninvasive Ventilation in COPD

Noninvasive ventilation refers to techniques that provide ventilatory support without endotracheal intubation [123]. Nowadays, the most frequently used form of NIV is positive-pressure ventilation delivered through a tight-fitting mask or helm, while negative-pressure support has a historical relevance and a narrow range of indications [124].

NIV is a widely used standard of care in an acute care setting as treatment for acute or acute-on-chronic hypercapnic respiratory failure [61]. Acute exacerbations (AE) of COPD are the most common indication of NIV as COPD patients hospitalized due to AE frequently develop acute respiratory failure (ARF) and acidosis [125]. The beneficial effects of NIV in ARF are robust and well proven; it reduces the need for endotracheal intubation, decreases the rate of complications, and lowers the cost of care [61,126]. Furthermore, NIV can also be used as long-term treatment for chronic hypercapnic respiratory failure in COPD [18,19].

### 6.1. NIV in Stable Hypercapnic COPD

Several studies have investigated the effects of long-term NIV (LT-NIV) on various outcomes in hypercapnic stable COPD, including mortality [93,127,128,129,130]; hospitalization rate [93,128,130,131]; and patient-centered outcomes, such as exercise tolerance [93,128,131,132,133,134,135], symptom severity [127,128,132,133,134,135,136], or health-related quality of life (HRQL) [93,128,129,132,134]. However, the results of these studies are controversial. The reasons for contradictions include differences in study design (observational or controlled trials), selection criteria (i.e., hypercapnic vs. normocapnic patients), patient population (frequent vs. infrequent exacerbator and inclusion vs. exclusion of obese patients or those with obstructive sleep apnea), or target PaCO_2_ (aiming to normalize hypercapnia or without a set PaCO_2_ target). Most importantly, the ventilator setups were different between studies. Increasing evidence suggests that appropriate intensity NIV that is able to normalize hypercapnia is the only method associated with any mortality benefit [93]. It has better efficacy in improving dyspnea, lung function, and quality of life compared to low-intensity NIV [137].

In 2014, Köhnlein et al. found that lowering carbon dioxide by at least 20% or to PaCO_2_ < 6.4 kPa significantly improved the one-year survival rate compared to standard therapy [93]. Furthermore, patients in the intervention group experienced a higher improvement in HRQL and several physiological parameters, such as pH, PaCO_2_, S_a_O_2_, HCO_3_^−^, and FEV_1_. Clini et al. found no difference in mortality between NIV and the control group with 2 years of follow-up, but the quality of life improved at a higher rate in the group receiving ventilatory support [128]. Additionally, when comparing the number of hospitalizations over the 2-year study period to the previous 3 years, they found that the hospitalization rate in the LT-NIV group decreased by 45%, while it increased by 27% in the control group. Casanova et al. did not find any difference in mortality, readmission rate, hospitalization, or endotracheal intubation rate between the NIV and the LTOT control group, but patients in the intervention arm were observed as having decreased dyspnea [127]. McEvoy et al. found a slight survival benefit in patients treated with NIV and LTOT compared to LTOT alone, but contrary to the previous studies, they reported reduced HRQL [129]. Márquez-Martín et al. assessed the effect of exercise training, NIV, and the combined use of these interventions on exercise capacity, quality of life, systemic inflammatory markers, gas exchange, peripheral muscle strength. and BODE index [135]. They reported a significant improvement in dyspnea, HRQL. and BODE index in all three treatment arms, with no statistical difference between the groups. Although there were differences in the other individual outcomes, the combination of NIV and exercise training were found to improve all parameters to a higher extent than each of the interventions separately. Duiverman et al. and Garrod et al. both evaluated whether the addition of NIV to a pulmonary rehabilitation program would produce greater benefits to HRQL and functional status than rehabilitation alone [134,138]. They observed greater improvements in HRQL scores (Chronic Respiratory Disease Questionnaire and Maugeri Respiratory Failure Questionnaire) and gas exchange parameters (PaCO_2_ and PaO_2_) in the group receiving NIV and rehabilitation than in the rehabilitation group alone.

Even though the results regarding primary outcomes of the studies are inconclusive, improved exercise tolerance, improved quality of life, and decreased dyspnea seem to be universal, which is also supported by observational studies [139,140]. This effect of NIV is paramount because, according to a recent large-scale international survey conducted by a European Respiratory Society (ERS) taskforce committee, these outcomes are the most important for patients living with severe COPD [141]. Based on these improvements in patient-centered outcomes and the fact that only minor adverse events occurred during the use of NIV, both the ERS and the American Thoracic Society (ATS) guidelines recommend the use of domiciliary NIV in chronic stable hypercapnic COPD [18,19].

### 6.2. NIV during Acute Exacerbations of COPD

Acute exacerbations are important events during COPD as they are a key cause of mortality, accelerate lung function decline, contribute to worsening quality of life, and enhance the risk of a future relapse [142,143]. A large proportion of patients with AECOPD admitted to hospital develop acute respiratory failure necessitating mechanical ventilation.

The ultimate treatment for life-threatening ARF is invasive mechanical ventilation. However, this is associated with a number of short- and long-term complications, such as ventilator-associated lung injury, laryngeal and tracheal injuries, nosocomial pneumonia, difficultly weaning, and prolonged ICU stay [144,145]. The large costs [146] and high risk of complications raises the need for the development of alternative therapies.

Noninvasive ventilation has been effectively used for decades for the acute treatment of ARF in AECOPD [147]. The review of the available literature would extend the scope of our article, and as it has been carried out on multiple instances by international review boards [17,61], we aim to summarize its benefits only briefly.

Several randomized controlled trials (RCT) have assessed the effectiveness of NIV vs. usual care in AECOPD patients experiencing an AHRF episode who do not initially require endotracheal intubation and mechanical ventilation. A meta-analysis of 17 RCTs involving more than 1200 patients found that NIV reduces the risk of mortality by 46% and the need for intubation by 65%. The quality of this evidence was graded “moderate” according to the GRADE criteria. The authors also concluded that NIV decreased the length of hospital stay and improved pH and blood gas parameters [17]. Similarly, in their joint guideline, the taskforce committees of the ERS and ATS also reported that, besides benefits in mortality and prevention of intubation, NIV shortens the length of intensive care unit (ICU) stay, decreases the rate of infectious complications, and improves dyspnea [61]. Based on the high certainty of the evidence, the ERS/ATS guideline strongly recommends the use of bilevel NIV for AECOPD patients with AHRF.

### 6.3. NIV after an Acute Exacerbation

In a proportion of patients surviving an AHRF episode due to AECOPD, hypercapnia persists after hospital discharge and even in stable state [125]. It raises the question whether these patients would benefit from long-term NIV therapy initiated right after a life-threatening episode of AE.

Cheung et al. carried out a pilot RCT involving patients surviving a life-threatening AE in which they compared the continuation of NIV after hospital discharge to a control sham continuous positive airway pressure (CPAP) therapy in hypercapnic patients [148]. They reported a significant difference in recurring ARF during the 1-year follow-up favoring NIV (38.5% in NIV group vs. 60.2% in control group, *p* = 0.039). However, the definition of AE was unclear, and the drop-out rate was high from an already limited sample size (25% of 49 patients). Notably, the LT-NIV or control therapy was initiated after >48 h of successful weaning of the original ventilation but before hospital discharge. De Backer et al. also conducted a pilot RCT comparing maximal pharmacological treatment to maximal pharmacological treatment combined with NIV therapy for 6 months after an episode of AE [149]. Although the sample size was very small (*n* = 15), they did not report any deaths or relapses during the study period. However, the primary outcomes of the study were not mortality or relapse rate but rather arterial blood gas values and functional imaging of the lungs. Importantly, they found that NIV provided better ventilation/perfusion match and thus improved the arterial blood gas and lung function parameters. Furthermore, two large-scale international RCTs also investigated the efficacy of LT-NIV + LTOT in reducing mortality and readmission rate after an AE compared to LTOT alone [150,151]. The RESCUE trial involving 201 hypercapnic COPD patients failed to detect any differences in the main outcomes between the two groups during the 1-year follow-up [151]. In contrast, the HOT-HMV trial found a remarkable difference in the 12-month risk of readmission or death (63.4% in the NIV group vs. 80.4% in the LTOT alone group) and in the median time to readmission or death (4.3 months vs. 1.4 months, respectively). This huge discrepancy may be explained by the different criteria of hypercapnia (>45 mmHg in RESCUE vs. >53 mmHg in the HOT-HMV trial) and by the different timepoints of initial assessment and enrolment of patients. In the RESCUE trial, the randomization was carried out after >48 h of independence from the ventilator, which may have led to the inclusion of patients who had a spontaneously reversible hypercapnia [152], thus masking the beneficial effects of NIV in the right population. In the HOT-HMV trial, however, the assessment of hypercapnia and the inclusion was carried out 2–4 weeks after the exacerbation.

Lastly, Funk and his colleagues carried out a study from a different approach [153]. They provided LT-NIV therapy to 26 consecutive COPD patients who remained hypercapnic after an episode of ARF, and after 6 months of therapy, they randomized the participants to either continue or withdraw from NIV. They observed that the risk of clinical worsening was higher in the withdrawal group, and the six-minute walk distance was reduced compared to the ventilation group. Table 1 describes these trials and summarizes their main findings.

As a consequence of the abovementioned evidence, the ATS and ERS recommend a careful selection of patients benefiting from LT-NIV therapy following an ARF, with the reassessment of hypercapnia 2–4 weeks after hospital discharge [18,19].

## 7. Discussion

### 7.1. Clinical Recommendations

Hypercapnia is common in patients with severe and very severe COPD, and it is a risk factor for hospitalization and mortality. Correction of PaCO_2_ values might therefore be beneficial in selected patients. Based on the available evidence, the authors make the following clinical recommendations (Table 2).

### 7.2. Future Research Directions

Previous studies have clearly highlighted that not all patients with hypercapnia benefit from long-term NIV treatment, and the effect is also variable. This could be due to the poor understanding about the role of hypercapnia on the course of COPD and also due to the fact that NIV trials did not systematically assess physiological and biological variables. Based on this, we make the following recommendations for future research directions:Observational studies are needed to understand the impact and stability of the hypercapnia exacerbator phenotype.Phase III long-term NIV trials with physiological and biological outcomes are needed in patients with chronic hypercapnic respiratory failure.Studies designed to understand the impact of long-term NIV on cardiovascular outcomes are needed.Studies designed to understand factors predicting treatment response to long-term NIV are needed.Studies designed to establish minimal hours of usage necessary with long-term NIV are needed.

## Figures and Tables

**Figure 1 jcm-11-03180-f001:**
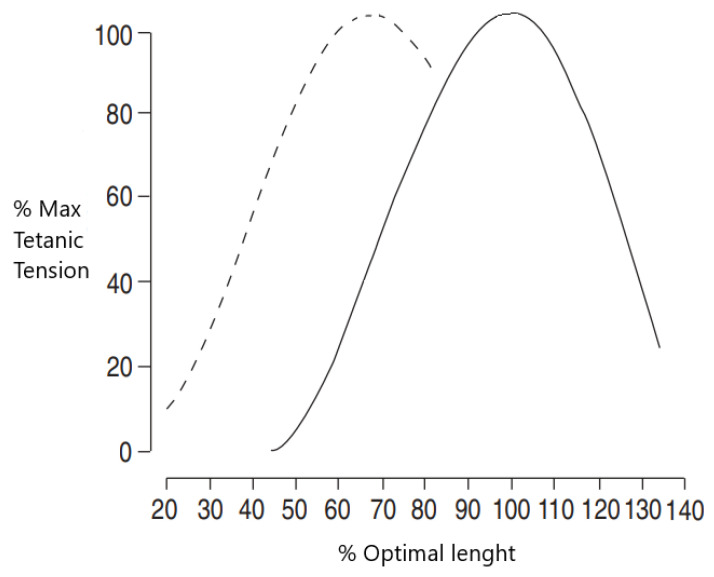
Diaphragmatic length–tension curve in subjects without (solid line) and with (dashed line) emphysema. In emphysema, the curve is shifted towards the left, thus shifting to shorter length.

**Figure 2 jcm-11-03180-f002:**
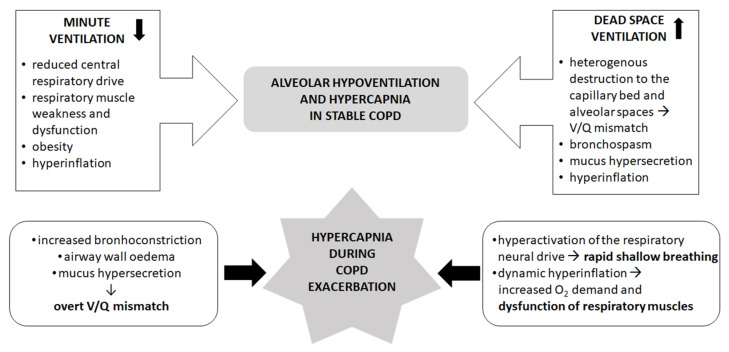
Mechanisms leading to hypercapnia in stable (**upper panel**) and exacerbated (**lower panel**) COPD. V/Q: ventilation/perfusion of the alveoli. References are listed in the text.

**Figure 3 jcm-11-03180-f003:**
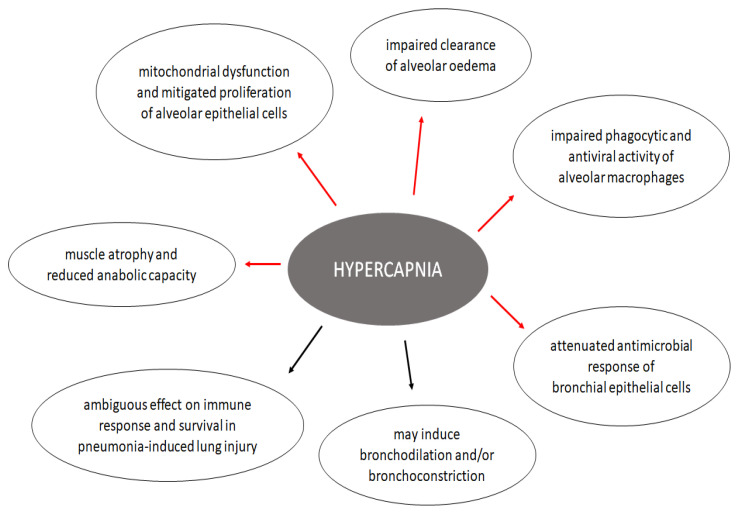
Possible cellular effects of hypercapnia in COPD. Red arrows indicate a harmful effect, while black arrows show that current evidence is inconclusive. References are listed in the text.

**Table 1 jcm-11-03180-t001:** Clinical trials investigating the effect of LT-NIV in chronic hypercapnic COPD on different outcomes.

Study Name	Population	Primary Outcome	Favors NIV	Baseline PaCO_2_, kPa	AE Frequency at Baseline	BMI	OSA	Normalizing Hypercapnia, Yes/No	NIV Mode
Casanova 2000 [127]	Stable	Number of AEs	No	6.8 ± 1.1	No data	25 ± 4	Excluded	No	Nasal BiPAP, S mode, EPAP: 4 cmH_2_O; IPAP: 12 cmH_2_O
Clini 2002 [128]	Stable	Arterial blood gas values, hospital and ICU admissions, total hospital and ICU length of stay, HRQL	Partly	7.2 ± 0.6	No data	26 ± 5	Excluded	Yes (5% decrease)	Nasal BiPAP, S/T mode, backup frequency: 8/min; EPAP: 2–5 cmH_2_O; IPAP: maximal tolerated pressure
Duiverman 2008 [134]	Stable	HRQL, functional status and gas exchange parameters	Yes	6.89 ± 0.68	No data	27.1 ± 6.4	Excluded	Yes (PaCO_2_ < 6.0 kPa)	BiPAP, S/T mode; IPAP: maximal tolerated pressure titrated towards an optimal correction of nocturnal arterial blood gases (PaCO_2_ 6.0 kPa and PaO_2_ 8.0 kPa)
Garrod 2000 [138]	Stable	Exercise capacity and health status	Yes	5.9 ± 0.9	No data	No data	Not excluded	No	Nasal BiPAP, S mode overnight or minimum 8 h/day, settings adjusted individually to obtain the maximal pressure tolerated; EPAP: 4 (4–6) cmH_2_O; IPAP: 16 (13–24) cmH_2_O
Köhnlein 2014 [93]	Stable	1-year all-cause mortality	Yes	7.8 ± 0.8	No data	24.8 ± 5.8	Not excluded	Yes (>20% decrease or PaCO_2_ < 6.5 kPa)	Pressure support ventilation with high backup rates minimum 6 h/day, preferably during sleep (face or nasal mask). Aim: to reduce ≥20% baseline PaCO_2_ or PaCO_2_ < 6.5 kPa
Marquez-Martin 2014 [135]	Stable	Exercise capacity	Favors ventilation/training combined group over ventilation alone	NIV group: median 51, NIV-ET group: median 50	No data	No data	Excluded	No	Nocturnal nasal BiPAP, S/T mode, backup frequency 12/min, 6–8 h/night; EPAP: 4 cmH_2_O; IPAP: initially 10 cmH_2_O and increased progressively to a maximum of 20 cmH_2_O, depending on patient tolerance, clinical response and SpO_2_
McEvoy 2009 [129]	Stable	Survival	Yes	7.01 [6.80–7.23]	No data	25.5 [24.3–26.7]	Excluded	No	BiPAP, VPAP mode, EPAP: lowest possible level (~3 cm H_2_O); IPAP: gradually increased during daytime and night-time trials to the maximum tolerated with a target PS of ≥10 cm H_2_O
Cheung 2010 [148]	Post AE (>48 h after successful weaning of acute NIV)	Recurrent severe AE with AHRF requiring acute NIV, intubation or resulting in death in the first year	Yes	7.7 ± 1.0	Previous acute NIV: 1 [0–3], previous intubation: 0 (0–1), no other data	19.2 ± 3.6	Excluded	No	BiPAP, S/T mode, backup frequency: 14/min; EPAP: 5 cmH_2_O; IPAP: 10–20 cmH_2_O
De Backer 2011 [149]	Post AE (5–12 days after admission)	Arterial blood gas values and functional imaging of the lungs	Yes	7.39 ± 1.03	No data	No data	Excluded	Yes (5% decrease)	BiPAP for >5 h a day with a full face mask; modes were adapted until O_2_ saturation was >90% during 90% of the time, and PaCO_2_ was decreased 5% in 1 h
Funk 2011 [153]	Post AE (before discharge from the ICU or immediately after transfer to regular wards)	Time to clinical worsening Defined as an escalation of mechanical ventilation	Yes	7.6 ± 1.7	No data	24.2 ± 4.3	Excluded	No	BiPAP EPAP: ~5 cmH_2_O; IPAP: increasingly raised from 10 to ~20 cmH_2_O. The inspiratory time was limited to a maximum of 1.3 s
Murphy 2017 [150]	Post AE (2–4 weeks after resolution of respiratory acidemia)	Time to readmission or death within 12 months adjusted for the number of previous COPD admissions, previous use of long-term oxygen, age, and BMI	Yes	7.87 ± 0.93	≥3 COPD-related readmissions within past year: NIV-LTOT group: N = 30 (53%) vs. LTOT group: N = 31 (53%)	21.5 (18.8–24.5)	Excluded	Yes (reduce tcCO_2_ by at least 4 mmHg)	BiPAP, PS mode, recommended initial titration settings: IPAP 18 cmH_2_O, EPAP 4 cmH_2_O, backup rate 14–16/min; target IPAP ≥25 cmH_2_O. NIV settings and O_2_ flow rate were titrated to maintain SpO_2_ >88% and to reduce tcCO_2_ by ≥4 mmHg
Struik 2014 [151]	Post AE (>48 h after termination of ventilatory support)	Time to readmission for respiratory cause or death	No	7.9 ± 1.2	Median: 2, min–max: 1–9	24.6 ± 5.4	Excluded	Yes (to achieve normocapnia)	BiPAP, S/T mode starting with a backup frequency of 12/min; IPAP: initial 14 cmH_2_O and gradually increased to a maximal tolerated level; EPAP: initial 4 cmH_2_O and increased if auto-PEEP was present or when patients used respiratory muscles to trigger the ventilator. Respiratory rate was set as close as possible to the that of the patient. I:E ratio was 1:3, with a short rise time and then titrated on comfort and effectiveness

Abbreviations: AE, acute exacerbation; AHRF, acute hypercapnic respiratory failure; BiPAP, bilevel positive airway pressure; BMI, body mass index; COPD, chronic obstructive pulmonary disease; EPAP, expiratory positive airway pressure; I:E, ratio of inhalation to exhalation; IPAP, inspiratory positive airway pressure; LT, long-term; NIV, noninvasive ventilation; OSA, obstructive sleep apnea syndrome; PaCO_2_, partial arterial carbon dioxide pressure; PEEP, positive end-expiratory pressure; PS, pressure support; SpO_2_, arterial oxygen saturation; S/T mode, spontaneous/timed mode.

**Table 2 jcm-11-03180-t002:** Clinical recommendations on the screening, assessment, and treatment of stable hypercapnic COPD.

Category	Recommendation
Screening	Patients with severe and very severe COPD and those on long-term oxygen therapy should have regular blood gas assessment.
	Patients with acute hypercapnic respiratory failure should have a blood gas assessment at 2–4 weeks following discharge.
Assessment	Pharmacological and nonpharmacological COPD treatment and other disorders causing hypercapnia (i.e., obesity, neuromuscular, and chest wall diseases) should be evaluated during assessment.
	Routine sleep study should be offered to explore the presence of obstructive sleep apnoea and to identify variable (i.e., sleep-phase or positional) episodes of hypoventilation.
Treatment	Pharmacological therapy should be optimised to improve symptoms and reduce the number of exacerbations.
	Treatable traits contributing to hypercapnia (i.e., obesity and sarcopenia) should be addressed in parallel with NIV.
	Long-term NIV should be offered to those with persistent hypercapnic respiratory failure (PaCO_2_ ≥ 52 mmHg (>6.8 kPa)).
	The effect of long-term NIV therapy should be assessed with routine blood gas tests, sleep studies, and COPD-related outcomes (i.e., symptoms, quality of life, and the number of exacerbations).
	NIV treatment should be titrated to normalise PaCO_2_ (PaCO_2_ < 52 mmHg (<6.8 kPa)).

## Data Availability

Not applicable.

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
