# Peer review of "Hypercapnia in COPD: Causes, Consequences, and Therapy"

_jcm, 2022, doi:10.3390/jcm11113180_

Round 1
Reviewer 1 Report
Summary: Drs Csoma et al have created an informative review of the etiology, therapy, and outcome in patients with hypercapnia. The authors cover the systemic effects of hypercapnia on various organ systems, endothelial function, pulmonary hypertension, and airway mechanics. It is well written and comprehensive with substantial references.
Major Issues:
None
Minor Issues:
1) page 1, line 37 consider using the word 'present' for 'already'
2) page 2, line 52. Define 'chronic type II respiratory failure'
3) page 2, lines 69-77 Consider adding the concept of 'cannot breath' for mechanical dysfunction of the respiratory pump, and 'won't breath' for central and neurologic forms of hypercapnia.
4) page 2, line 85. Consider adding an illustration demonstrating the 'unfavorable length-tension curve'
5) page 3, line 90 'at' should probably be 'during'
6) page 3, line 119. Consider if obesity may in part be 'compensatory' for abnormal COPD respiratory mechanics that result in hyperinflation.
7) Page 9, line 342. Is normalizing pH or pCO2 the important factor in these chronically stable hypercapnic patients in terms of outcomes (morbidity & mortality)
8) page 10, line 387 I believe 'ultima ratio for the' should be 'ultimate treatment'
9) Table 1. Consider moving this to an appendix and summarizing the findings in the text.
10) page 15, line 464. Consider a header for this section something like 'Necessary future research directions'
6)
Author Response
Comment: page 1, line 37 consider using the word 'present' for 'already'
Response: Thank you for the comment. We corrected this.
Comment: page 2, line 52. Define 'chronic type II respiratory failure'
Response: Thank you. This is explained in the revised manuscript.
Comment: page 2, lines 69-77 Consider adding the concept of 'cannot breath' for mechanical dysfunction of the respiratory pump, and 'won't breath' for central and neurologic forms of hypercapnia.
Response: We have introduced these terms as requested.
Comment: page 2, line 85. Consider adding an illustration demonstrating the 'unfavorable length-tension curve'
Response: Thank you, a new figure has been included as requested.
Comment: page 3, line 90 'at' should probably be 'during'
Response: We have corrected this.
Comment: page 3, line 119. Consider if obesity may in part be 'compensatory' for abnormal COPD respiratory mechanics that result in hyperinflation.
Response: We have added this information, as requested.
Comment: Page 9, line 342. Is normalizing pH or pCO2 the important factor in these chronically stable hypercapnic patients in terms of outcomes (morbidity & mortality)
Response: Thank you for the comment. In the referenced part of the manuscript, we discuss studies which titrated NIV settings to normalise pCO2 rather than pH. Patients in these studies were included in their stable state which is naturally associated with normal pH. Of note, in the study where patients were included during their exacerbation (i.e. RESCUE trial), no benefit was noticed in morbidity or mortality in the LT-NIV group.
Comment: page 10, line 387 I believe 'ultima ratio for the' should be 'ultimate treatment'
Response: Thank you. We have corrected it.
Comment: Table 1. Consider moving this to an appendix and summarizing the findings in the text.
Response: Thank you. We believe that Table 1 is an important part of the paper as it summarises the main clinical trials and the reader can more easily compare them than reading the text. Readers less likely read appendix than the main article, therefore we decided to keep the Table 1 in the main manuscript.
Comment: page 15, line 464. Consider a header for this section something like 'Necessary future research directions'
Response: We have added headers to the two main parts of the Discussion.
Reviewer 2 Report
General comments:
The authors have a physiological, molecular biological, and clinical review of hypercapnia in COPD patients. It is necessary and sufficient, and reflects the results of recent research. There are no typographical errors and it is considered to be a worthy publication.
Major comments: None
Minor comments: None
Author Response
We really appreciate your kind comments.